# Transcriptomics and Immunological Analyses Reveal a Pro-Angiogenic and Anti-Inflammatory Phenotype for Decidual Endothelial Cells

**DOI:** 10.3390/ijms20071604

**Published:** 2019-03-31

**Authors:** Chiara Agostinis, Elisa Masat, Fleur Bossi, Giuseppe Ricci, Renzo Menegazzi, Letizia Lombardelli, Gabriella Zito, Alessandro Mangogna, Massimo Degan, Valter Gattei, Marie-Pierre Piccinni, Uday Kishore, Roberta Bulla

**Affiliations:** 1Institute for Maternal and Child Health, Istituto di Ricovero e Cura a Carattere (I.R.C.C.S.) “Burlo Garofolo”, via dell’Istria 65/1, 34134 Trieste, Italy; cagostinis@units.it (C.A.); fbossi@units.it (F.B.); giuseppe.ricci@burlo.trieste.it (G.R.); gabriella.zito@burlo.trieste.it (G.Z.); 2Department of Life Sciences, University of Trieste, via Valerio 28, 34127 Trieste, Italy; elisa.masat@gmail.com (E.M.); menegazz@units.it (R.M.); 3Department of Medical, Surgical and Health Science, University of Trieste, 34127 Trieste, Italy; alessandro.mangogna@gmail.com; 4Department of Experimental and Clinical Medicine, Internal Medicine Unit, University of Florence, Largo Brambilla 3, 50134 Florence, Italy; letizialombardelli@gmail.com (L.L.); mppiccinni@hotmail.com (M.-P.P.); 5Clinical and Experimental Onco-Hematology Unit, Centro di Riferimento Oncologico, I.R.C.C.S., via Gallini 2, 33081 Aviano (PN), Italy; mdegan@cro.it (M.D.); vgattei@cro.it (V.G.); 6Biosciences, College of Health and Life Sciences, Brunel University London, Uxbridge UB8 3PH, UK; uday.kishore@brunel.ac.uk

**Keywords:** endothelium, decidua, skin, angiogenesis, inflammation

## Abstract

Background: In pregnancy, excessive inflammation and break down of immunologic tolerance can contribute to miscarriage. Endothelial cells (ECs) are able to orchestrate the inflammatory processes by secreting pro-inflammatory mediators and bactericidal factors by modulating leakiness and leukocyte trafficking, via the expression of adhesion molecules and chemokines. The aim of this study was to analyse the differences in the phenotype between microvascular ECs isolated from decidua (DECs) and ECs isolated from human skin (ADMECs). Methods: DECs and ADMECs were characterized for their basal expression of angiogenic factors and adhesion molecules. A range of immunological responses was evaluated, such as vessel leakage, reactive oxygen species (ROS) production in response to TNF-α stimulation, adhesion molecules expression and leukocyte migration in response to TNF-α and IFN-γ stimulation. Results: DECs produced higher levels of HGF, VEGF-A and IGFBP3 compared to ADMECs. DECs expressed adhesion molecules, ICAM-2 and ICAM-3, and a mild response to TNF-α was observed. Finally, DECs produced high levels of CXCL9/MIG and CXCL10/IP-10 in response to IFN-γ and selectively recruited Treg lymphocytes. Conclusion: DEC phenotype differs considerably from that of ADMECs, suggesting that DECs may play an active role in the control of immune response and angiogenesis at the foetal-maternal interface.

## 1. Introduction

Endothelial cells (ECs) form a continuous barrier between blood and tissues and act as a gateway to the traffic of molecules and cells across the vessel wall, playing an active role in homeostasis, inflammation and immunity [1]. They also play an important role in the onset and maintenance of inflammation, being both target as well as source of cytokines, chemokines and growth factors [1]. ECs, although apparently similar to each other in function and morphology, represent a heterogeneous population of cells in terms of inflammatory mediators secretion, adhesion molecules modulation, leakiness and pro-coagulant activity [2]. The pathophysiological functions of ECs, which mainly relates to angiogenesis and inflammation, take place at the level of the microvascular bed, which constitutes the bulk of the overall endothelial surface [3]. The microenvironment as well as the epigenetics are involved in determining ECs heterogeneity, resulting in tissue-specific properties [4]. Skin is the primary interface between the body and the environment; skin injury or infections promptly result in the activation of an efficient inflammatory response. Dermal microvascular ECs are active regulators of the inflammatory process [5]. They contribute to the secretion of inflammatory mediators, bactericidal molecules, increasing leakiness and pro-coagulant activity, modulation of adhesion and transmigration of leukocytes through the expression of adhesion molecules and chemokines [6].

Decidua, the maternal part of human placenta, is derived from uterine endometrium through an inflammatory process that leads to the transformation of endometrial stromal cells [7]. Consequently, decidualization contributes to the activation of the local immune cells, such as Natural Killers (NK), macrophages and T lymphocytes. These immune cells play a dual role in the decidual environment, establishing immune tolerance and a mild inflammatory milieu, both being important for implantation and pregnancy outcome [8]. During the early stages of pregnancy, a state of mild systemic inflammation at the foetal-maternal interface is revealed by the presence of an activated vascular endothelium, leukocytosis and increased activation of immune cells such as monocytes. Increased plasma levels of inflammatory cytokines and chemokines, such as IL-8, IL-18, IL-12 and TNF-α are also observed [9]. Moreover, several immune cells, mainly uterine NK cells and macrophages, and to a lesser extent, dendritic cells and T cells, are diffusely embedded in the decidual microenvironment and contribute to the control of the immune response [10].

Decidual endothelial cells (DECs) are influenced by this special immunological and inflammatory environment established in the decidua and play a key role in controlling the traffic of leukocytes across the vessel wall, which needs to be tightly regulated in order to guarantee the success of the pregnancy. In addition, DECs are involved in angiogenesis, a necessary process for decidualization, which in turn, is an essential step in the maturation of new blood vessels in mammalian pregnancy.

The aim of this study was to characterize the phenotype of DECs in terms of expression of molecules contributing to angiogenesis and leukocyte recruitment, and to compare it with the microvascular ECs phenotype from human skin, which is normally geared to mount aninflammatory response. Adult Dermal Microvascular ECs (ADMECs) were chosen as a cellular model representingmicrovascular ECs that are physiologically involved in inflammatory responses. Although similar to the commonly used HUVEC, ADMECs are more responsive to inflammatory stimuli [11] and express different levels of endothelial markers [12].

## 2. Results

### 2.1. Decidual Endothelial Cells Express a Different Profile of Angiogenic Factors and Adhesion Molecules Compared to Endothelial Cell Isolated from Dermal Skin

Since DECs are the only ECs, which under physiological conditions, are able to synthesize the complement component C1q [13], an extensive characterization of DEC phenotype was carried out. Initially, a gene expression profiling using isolated primary human ADMECs and DECs was performed. The isolated cells were found to be 100% positive for the pan-endothelial cell markers VE-cadherin (CD144), von Willebrand Factor (vWF), and mesenchymal marker, vimentin, as assessed by immunostaining (Figure 1). A comparison of the gene expression profiles between DECs and ADMECs revealed 1909 transcripts that were differentially expressed (*t*-test *p*-value <0.01 and fold change ≥1), 1652 of which were up-regulated and 255 down-regulated in the DEC population (Appendix A).

Analysis of the gene expression profile (GEP) data revealed, besides other genes, interesting differences in the expression of genes involved in angiogenesis and leukocyte recruitment. In particular, we observed a 2.7 fold up-regulation of VEGF-A; a 2.1 fold increase in the expression of HGF, and even more pronounced up-regulation (11.6 fold) of IGFBP3 (Appendix A). The expressions of VEGF-A, HGF and IGFBP3 transcripts were validated by RT-qPCR using microarray samples as well as five additional EC populations each obtained from other subjects (Figure 1B). The results presented in Figure 1B confirmed the difference in gene expression levels of these growth factors. The production of VEGF-A, HGF and IGFBP3 was also analysed at protein level by ELISA in the supernatants of five different populations of DECs and ADMECs (Figure 1C). The results shown in Figure 1C indicated that DECs produced higher amount of VEGF-A, IGFBP3 and HGF as compared to ADMECs. The microarray and RT-qPCR analysis also revealed a higher expression of the constitutive adhesion molecules, ICAM-2 and ICAM-3, by DECs. These data were confirmed at the protein level by cytofluorimetric analyses (Figure 1D).

### 2.2. DECs Respond Feebly to Vasoactive Stimuli Leading to Vascular Leakage Compared to ADMECs

To investigate the unique characteristics of DECs in terms of inflammatory response, an endothelial permeability assay was carried out. ECs were grown to confluence onto a transwell (TW) insert. To measure the vascular permeability, ECs were treated with Platelet-Activating Factor (PAF), Histamine (HIS) or Bradykinin (BK). To determine the increase in ECs permeability, FITC-conjugated BSA was added to the upper chamber. The passage of FITC-BSA to the lower chamber was evaluated after 5, 15 and 30 min and the results were expressed as % of BSA leakage. As shown in Figure 2, ADMECs promptly responded to all three vasoactive stimuli. On the contrary, DECs were not responsive to vasoactive stimuli even after 30 min of incubation.

### 2.3. DECs Produce Lower Levels of Oxygen-Derived Reactive Molecules than ADMECs in Response to TNF-α and Histamine Challenge

We evaluated the capability of DECs and ADMECs to produce H_2_O_2_, an oxygen-derived reactive molecule endowed with bactericidal properties. Both ECs, cultured in 96 well plates, were incubated with Ampliflu Red reagent, a molecule that produces a fluorescent signal following its reaction with H_2_O_2_. As expected, ADMECs were able to produce high levels of H_2_O_2_ in response to both TNF-α and histamine stimulation. On the contrary, DECs maintained a lower level of H_2_O_2_ production following stimulation with TNF-α and histamine (Figure 3).

### 2.4. DECs, Compared to ADMECs, Are Weak Responders to TNF-α Stimulation with Respect to Chemokine Secretion, Adhesion Molecule Expression and Leukocyte Recruitment

Another important aspect of ECs function is their key role in controlling leukocyte trafficking across the vessel wall. It is well known that ECs, including ADMECs, promptly respond to TNF-α by enhancing adhesion molecule expression and cytokine secretion [14,15]. This results in leukocyte adhesion and transmigration into tissue spaces. We, therefore, analysed the expression of selected chemokines (IL-8, MCP-1, MIP-1α and RANTES) in resting as well as TNF-α stimulated ECs. ECs were incubated overnight with TNF-α and supernatants were collected. ECs were then lysed and the total mRNA was extracted. The gene expression analysis (data not shown) and the chemokine quantification in the culture supernatant revealed a statistically significant weaker responsiveness of DECs to TNF-α stimulation (Figure 4A–D); only IL-8 production was comparable between DECs and ADMECs.

The expression of ICAM-1, VCAM-1 and E-Selectin, which are inducible adhesion molecules involved in leukocyte recruitment, can be modulated by TNF-α. Monolayers of DECs and ADMECs were stimulated with TNF-α for 4 h (or 18 h for VCAM-1) and the expression of these adhesion molecules was evaluated by ELISA on the whole cells. The results are shown in Figure 4E and are expressed as fold of increase compared to resting condition. ADMECs, as expected, showed a significant up-regulation of ICAM-1, VCAM-1 and E-Selectin expression on their cell surface after stimulation with TNF-α, whereas DECs showed weaker surface expression of adhesion molecules.

In order to ascertain if feeble surface expression of adhesion molecules by DECs, following TNF-α challenge, coincided with a functional deficit in the leucocyte recruitment, we performed a trans-endothelial migration assay using Lympho-Monocytes (LM) isolated from peripheral blood. ECs were grown to confluence in the TW system and stimulated with TNF-α. Subsequently, peripheral blood LM were added to the upper chamber of the TW, allowed to migrate for 30 min, and then counted. As showed in Figure 4F, TNF-α stimulated DECs brought about a considerably reduced LM recruitment compared to ADMECs.

### 2.5. IFN-γ-stimulated DECs Are Potent Recruiters of Natural Killer and Regulatory T Cells

Chemokines such as CXCL9/MIG and CXCL10/IP-10 are important in controlling the recruitment and/or retention of NK cells in human decidua [16]. Since CXCL9/MIG and CXCL10/IP-10 are inducible by IFN-γ, we asked whether the production of these two chemokines by DECs could be modulated in response to IFN-γ. As shown in Figure 5, IFN-γ-stimulated DECs, but not ADMECs, secreted high amounts of both chemokines.

We performed a migration assay with lympho-monocytes (LM) using the supernatant of cells stimulated with IFN-γ (as chemoattractant) in order to establish the functional consequences of these chemokines produced by DECs (Appendix A). In addition, we characterized the DEC-recruited cells that migrated to the lower compartment of the Boyden Chamber during the trans-endothelial migration assay. ECs were grown to confluence in the TW system and stimulated overnight with IFN-γ. Peripheral blood LM were then added in the upper chamber and allowed to migrate for 30 min. Migrated cells were stained for CD45^+^CD56^+^, CD3^+^CD4^+^, CD3^+^CD8^+^, and CD3^+^FoxP3^+^ using appropriate monoclonal Abs. Interestingly, besides CD45^+^CD56^+^ NK cells (Appendix A), DECs also exhibited a considerably enhanced ability to recruit Treg cells (CD3^+^FoxP3^+^), as compared to ADMEC (Figure 5C,D).

## 3. Discussion

In this paper, we report, for the first time, a detailed characterization of the phenotype of DECs that can differentially impact upon angiogenesis and leukocyte recruitment. As a point of reference, we used microvascular ECs derived from human skin which are normally robust participants during pro-inflammatory response.

Microvascular endothelial cells are active participants in and regulators of the inflammatory processes. They contribute to the process by (i) secreting inflammatory mediators and bactericidal molecules; (ii) increasing leakiness and pro-coagulant activity; (iii) modulating adhesion and migration of leukocytes through the expression of adhesion molecules and chemokines; and (iv) promoting pro-inflammatory cytokine expression [6].

In this study, we compared a range of immunological features of DECs and ADMECs, two types of microvascular ECs derived from very distinct anatomical sites. ADMECs are derived from a vascular location actively involved in the inflammatory process [5], whereas DECs are derived from decidua, where uncontrolled or heightened inflammatory milieu is unwarranted and can potentially lead to miscarriage or pregnancy related complications [17].

There are few studies concerning the characterization of DECs. Burrows and colleagues in 1994 described the basal level expression of adhesion molecules by immunohistochemistry, demonstrating that ICAM-1 is expressed by all vascular endothelium throughout the decidua [18]. However, it principally focused on differential marker expression ex vivo [18,19]. Alternatively, cells isolated from non-pregnant uterus were also employed to describe DEC properties [20,21,22].

We have previously shown that DECs are the only ECs able to express the complement protein C1q under basal physiologic conditions [13]. C1q, besides its well-known function as the recognition subcomponent of the complement classical pathway, is involved in trophoblast endovascular invasion [23]. The current study highlights a range of additional features of DECs. Most crucially, DECs produce a high amount of several important regulators of angiogenesis such as VEGF-A, HGF and IGFBP3. DECs also express high levels of the adhesion molecule ICAM-2 on their cell surface under basal conditions, and are the exclusive endothelial cell population that express ICAM-3, characteristics that have only been associated with the High Endothelial Venules (HEVs) so far [24].

DECs are unable to enhance inflammasome components in response to LPS stimulation [25]. More recently, comparing their immunological phenotype to that of other ECs such as HUVEC and ADMEC, DECs were found to be hypo-responsive to LPS stimulation in terms of IL-6, CXCL8/IL-8 and CCL2/MCP-1 production since they expressed low levels of TLR4 and manifested strong constitutive activation of the non-canonical NF-κB pathway and low responsiveness of the canonical pathway to LPS [26].

ECs play a pivotal role in regulating the selective passage of molecules that need to be recruited at the extravascular sites and participate in the control of fluid movement between luminal and ab-luminal sides of the vessels. Vascular permeability increase takes place mainly in post-capillary venules where the ECs are able to respond to different stimuli such as histamine, thrombin and bradykinin [27,28]. Indeed, ADMEC showed a prompt permeabilizing response to all the stimuli used (PAF, HIS, BK). DECs, however, were unable to show an increased vascular permeability. This unusual behaviour probably reflects some differences in receptor expression or intracellular signals triggered by receptor-agonist interaction. For BK, there are different types of receptors expressed on the cell surface and it has been demonstrated that only B2, and not B1, receptors are expressed at decidual level [29]. Moreover, as compared to other situations that involve specific modulation of vascular leakage, it has been shown that B1 receptor could also be important in the dermal vascular permeability [30], indicating a differential functional response between DEC and ADMEC. Besides vascular permeability, DECs are less efficient in the production of bactericidal molecules both in basal condition and in response to TNF-α and histamine stimulation.

Interestingly, while analysing DECs responses, an absent or mild modulation of the adhesion molecules, ICAM-1, VCAM-1 and E-selectin respectively, was observed when compared to ADMEC. Low or undetectable amounts of the chemokines, CCL2/MCP-1, CCL3/MIP-1α and CCL5/RANTES, after stimulation with the pro-inflammatory cytokine TNF-α, were found in DECs. This is in contrast to the typical effect of TNF-α that normally induces endothelial activation, which is accompanied by an increase in adhesion molecule expression and cytokine production [31]. These data indicate that DECs are hyporesponsive to pro-inflammatory stimuli in comparison with other endothelial cells and could actively participate in the control of leukocyte infiltration/trafficking in decidua.

DECs appear to be unique in their ability to respond to IFN-γ, another pro-inflammatory cytokine, mainly produced in decidua by decidual NK cells [32]. DECs, but not ADMECs, produced a large amount of CXCL9/MIG and CXCL10/IP-10. DECs were shown to be able to express the mRNA for CXCL10/IP-10 also after progesterone treatment [33] and in response to trophoblast conditioned medium exposure [34]. The production of CXCL10/IP-10 was shown to be important for the recruitment of uterine NK cells in maternal decidua inducing the migration of peripheral blood NK cells derived from first trimester pregnant women, across DECs and decidual stromal cells [33]. In this paper, we also demonstrated that DECs stimulated with IFN-γ recruited a higher percentage of CD3^+^FoxP3^+^ cells in comparison to ADMECs, although other markers (CD25, CD127) will be useful for a further characterization of Treg. Whether dysfunctional decidual endothelium can tip the balance away from the preponderance of Treg needs to be ascertained since maternal Tregs have been linked with a number of pregnancy-related complications. It is clear from this study that DECs are likely to be one of the lynchpins for proper recruitment and functioning of maternal NK cells, decidual macrophages and maternal Treg, each cell type being known to play a hierarchical role within the three trimesters of pregnancy.

## 4. Materials and Methods

### 4.1. Cell Isolation and Culture

Decidual first trimester biopsy specimens were obtained from healthy women (*n* = 8) undergoing voluntary termination of pregnancy at 8–12 weeks of gestation. Skin samples were obtained from women (*n* = 6) of fertile age undergoing reductive plastic surgery. The study was approved by the institutional review board of the Institute for Maternal and Child Health (IRCCS “Burlo Garofolo”, Trieste, Italy. Approval 13 December 2009). I Informed consent was obtained from all women providing the tissue specimens. DECs and ADMECs were isolated and characterized as previously described [13]. Each population corresponded to a single woman. Both ECs were seeded in a 12.5 cm^2^ flask precoated with 2 µg/cm^2^ fibronectin (Roche, Milan, Italy) and maintained in endothelial serum free basal medium (Life Technologies, Monza, Italy), supplemented with 20 ng/mL basic Fibroblast Growth Factor, 10 ng/mL Epidermal Growth Factor (Life Technologies), 10% *v*/*v* foetal calf serum (FCS) (Life Technologies), and 10% *v*/*v* human serum.

DECs were positively selected using Dynabeads M-450 (Life Technologies) coated with Ulex europaeus 1 lectin (Sigma–Aldrich, Milan, Italy), whereas ADMECs were further purified from a subconfluent mixed cell population using CD31-conjugated magnetic beads from Dynabeads (Life Technologies). When the cells were seeded for the experiments, the medium used was the same as described above but without human serum.

### 4.2. Immunofluorescence

ECs were plated on 8-chamber culture slides (BD Biosciences Discovery Labware, Milan, Italy) coated with 2 µg/cm^2^ fibronectin (Roche, Milan, Italy). When cells grew to confluence, they were fixed and permeabilized with FIX & PERM (Società Italiana Chimici, Rome, Italy). Then cells were incubated with primary monoclonal antibody (mAb) (clone 9) mouse anti-human vimentin (Sigma-Aldrich), (cloneF8/86) mouse anti-human vWF (Dako-Cytomation, Milan, Italy), or mouse anti-human VE-Cadherin (kindly provided by Prof. Dejana from Institute of Molecular Oncology, Milan, Italy) (5 µg/mL) for 1 h at room temperature (RT) followed by fluorescein isothiocyanate (FITC)-conjugated goat anti-mouse IgG for 1 h at RT. Images were acquired with Leica DM3000 microscope (Leica, Wetzlar, Germany) and collated using a Leica DFC320 digital camera (Leica).

### 4.3. RNA Isolation, cDNA Synthesis and Quantitative Real-Time Polymerase-Chain Reaction (RT-qPCR)

RNA was extracted from cells with euroGOLDtrifast (Euroclone, Milan, Italy) according to the supplier’s instructions and reverse transcripted as previously described [13]. qPCR was carried out on a Rotor-Gene 6000 (Corbett, Explera, Ancona, Italy) using iQ SYBR Green Supermix (Bio-Rad, Milan, Italy). Appendix A shows the primers used for RT-qPCR. The melting curve was recorded between 55 °C and 99 °C with a hold every 2s. The relative amount of gene expression in each sample was determined by the Comparative Quantification (CQ) method supplied as part of the Rotor Gene 1.7 software (Corbett Research) [35]. The relative amount of each gene was normalized with 18S and expressed as arbitrary units (AU) considering 1 AU obtained from decidual tissue used as a calibrator.

### 4.4. Microarray

Total RNAs from ECs were validated for integrity and purity using the Agilent 2100 Bioanalyzer (Agilent Technologies, Santa Clara, CA, USA). Single-color hybridization microarray experiments for gene expression profile (GEP) were performed with 100 ng total RNA/sample labeled with Cyanine (Cy)-3 dye using the Low Input Quick Amp Labeling Kit (Agilent Technologies). Cy3-labeled RNA was hybridized to the Whole Human Genome (4 × 44 K) oligo microarray platform (Agilent Technologies). Microarray slides were analysed with an Agilent Microarray Scanner (Agilent Technologies). The hybridization signal values for the multiple probes for each gene were obtained with the use of Agilent Feature Extraction Software 10.7.3 (Agilent Technologies). Bioinformatics analyses were performed using GeneSpring 11.5.1 (Agilent Technologies). GEP results were visualized by hierarchical clustering, applying Ward’s method with Euclidean distance [36]. GEO number: GSE41946.

### 4.5. Growth Factors and Chemokines Detection

ECs were grown to confluence in 24 wells plate (BD Falcon) in serum-free medium and stimulated for 24 h with IFN-γ (100 U/mL) or TNF-α (100 ng/mL) (Peprotech, Milan, Italy). The levels of VEGF-A, HGF, IGFBP3 were measured by commercial ELISAs following the manufacturer’s instruction (VEGF Human ELISA Kit, Invitrogen Milan Italy; Boster Immunoleader Tema Ricerca, Bologna Italy, for detection of HGF; DRG Products Tema Ricerca, Bologna Italy for detection of IGFBP3). The quantitative determination of IL-8, MCP-1, MIP-1α, RANTES, CXCL9/MIG and CXCL10/IP-10 was performed by a bead-based multiplex immunoassay (Biorad) and a Bioplex 200 system (Biorad Laboratories, Hercules, CA, USA), as previously described [37]. The cells were then lysed for the quantification of the total protein content by Bradford assay [24].

### 4.6. Migration Assays

LM were isolated as previously described [24]. ECs (2 × 10^4^) were seeded onto 20 μg/mL fibronectin-coated polycarbonate insert of a 24-well TW system (6.5 mm diameter, 8-μm pores; Corning Costar, Milan, Italy) and used 5 days after culture as previously described [33]. LM suspension (2 × 10^5^ cells/100 μL endothelial serum free basal medium with 0.1% *w*/*v* bovine serum albumin) was added to the upper compartment of the TW in the presence or absence of ECs and allowed to migrate through DECs for 1 h at 37 °C. Amounts of 10% FCS, IFN-γ, or conditioned medium (CM) of DECs treated with IFN-γ were added to the lower chamber as a chemoattractant for LM. The number of cells transmigrated was evaluated by Coulter Counter (Coulter Electronics, Luton, UK).

Migrated LM were fixed with FIX & PERM^®^ cell fixation and permeabilization kit (Società Italiana Chimici), according to the manufacturer’s instructions. A total number of 5 × 10^5^ cells were incubated on shaking at 800 rpm overnight at 4 °C with PE- or FITC-conjugated mouse monoclonal antibody (mAb) specific to human CD3 (clone MEM-57), CD8 (clone MEM-31), CD45 (clone MEM-28), CD56 (clone MEM-188) (ImmunoTools GmbH) and IgG1 and IgG2a (ImmunoTools GmbH) isotype controls. Cells were fixed with 1% *v*/*v* PFA (Sigma-Aldrich) and analysed for fluorescence with a FACSCalibur flow cytometer (BD Falcon) using CellQuest software (version 5.1, Becton Dickinson European HQ, Erembodegem-Aalst, Belgium).

### 4.7. Detection of Adhesion Molecules on ECs

ICAM-2 and ICAM-3 were examined via cytofluorimetric analysis [13]. ECs were incubated for 1 h at 37 °C with the phycoerythrin (PE)-conjugated mAb ICAM-2 (clone CBR-IC2/2) or with PE-conjugated mAb ICAM-3 (clone CBR-IC3/1) (Biolegend, Milan, Italy) and their relatives PE-conjugated mouse IgG2a or IgG1 (ImmunoTools GmbH, Friesoythe, Germany) isotype controls. Quantization of ICAM-1, VCAM-1 and E-Selectin was evaluated by an ELISA on the whole cells, in ECs stimulated for 4 h (for ICAM-1 and E-Selectin) or 18 h (for VCAM-1) with TNF-α (100 ng/mL) and then incubated with monoclonal primary antibodies (mAb 6.5B5 anti-ICAM-1, mAb clone 1.4C3 anti-VCAM-1 Sigma-Aldrich or mAb clone 1.2B6 anti-E-selectin; 5 µg/mL) for 1 h at RT, as previously described [38]. The cells were then lysed for the quantification of the total protein concentration via Bradford assay [24].

### 4.8. Endothelial Leakage

Human endothelial cells (ECs; 2 × 10^4^) were seeded onto polycarbonate inserts of a 24-well TW system (6.5-mm diameter, 3-µm pores; Corning Costar, Milan, Italy) coated with 2 µg/cm^2^ of human fibronectin and used after reaching the confluence. Each TW was checked for the formation of intact monolayer on the insert by adding FITC-BSA (1 mg/mL) to the upper chamber and measuring the amount of labelled BSA that passed down to the lower chamber by an Infinite200 (TECAN). The TWs were used only when the intensity of fluorescence in the lower chamber was negligible; in this case, the stimuli were added to the upper chamber together with FITC-BSA and the fluorescence evaluated in the lower chamber at various time points.

### 4.9. Measurement of Total H_2_O_2_ Production

Hydrogen peroxide (H_2_O_2_) production was measured using Ampliflu Red (Sigma-Aldrich) reagent. ECs were seeded on 96-well plate to reach 90% confluence. To assess total H_2_O_2_, the cell culture medium was substituted with PBS + 2% *w*/*v* BSA + 0.7 mM MgCl_2_ and 0.7 mM CaCl_2_ containing 40 µM Ampliflu Red reagent, 1 µg/mL HRP, 5 µg/mL SOD and 100 µM NaN_3_ in a final volume of 100 µL. After 5 min of preincubation with TNF-α (100 ng/mL) or Histamine 10^−5^ M, the readings were taken at 576 nm with Infinite200 (TECAN). The cells were then lysed for the quantification of the total protein concentration by Bradford assay [24].

### 4.10. Statistic Analysis

Data were analysed using Two-way ANOVA, Tukey–Kramer test, and unpaired two-tailed Student’s *t*-test or one-way ANOVA with Bonferroni corrections. Results were represented as mean ± SEM. Non-parametric data were assessed by Mann–Whitney U tests and the results were expressed as median and interquartile range. *p* values  < 0.05 were considered significant. All statistical analyses were performed using Prism 6 software (GraphPad Software Inc., La Jolla, CA, USA).

## 5. Conclusions

DECs differ from ADMECs with respect to the production of several growth factors that play important roles in the control of angiogenesis. DECs also differ from ADMECs in the expression of adhesion molecules and the release of chemokines involved in local leukocyte recruitment, thus contributing to the establishment of the special decidual microenvironment. Our results indicate that DECs display a differential “arsenal” of adhesion molecules and cytokines in response to classical pro-inflammatory stimuli compared to ADMECs and HUVEC: they express constitutively the adhesion molecules ICAM-2 and ICAM-3, but fail to show increased expression of ICAM-1. These cells produce higher levels of the chemokines CXCL9/MIG and CXCL10/IP-10 in response to IFN-γ compared to other ECs, and promote selective migration of FoxP3 positive T cells. These findings are consistent with the local changes that occur during pregnancy which are designed to control the inflammatory response at the foetal-maternal interface. Thus, similar to T-cells, macrophages, dendritic cells and neutrophils, there may be a paradigm shift in the endothelial cells towards an anti-inflammatory phenotype EC2.

## Figures and Tables

**Figure 1 ijms-20-01604-f001:**
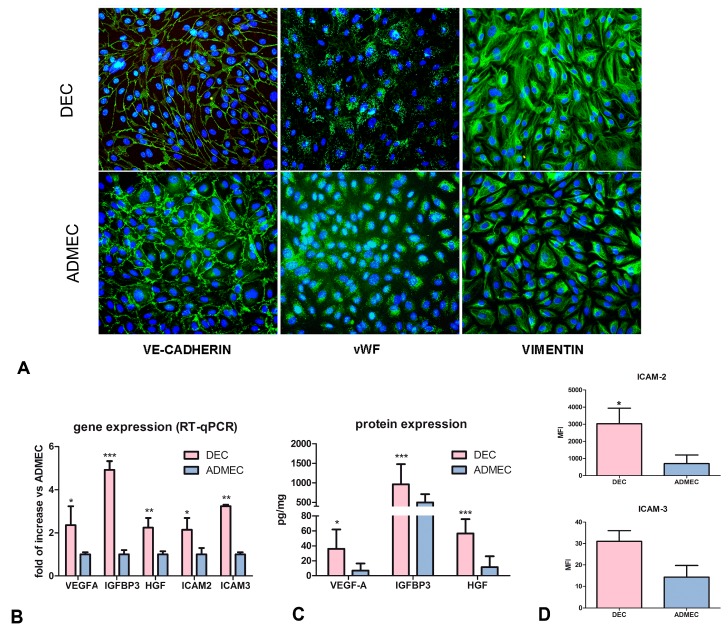
Phenotypic characterization of DECs and ADMECs. (**A**) Immunofluorescence analysis of vWF, VE-cadherin and vimentin on isolated and cultured DECs and ADMECs. Original magnification: 200×. (**B**) RT-qPCR of VEGF-A, HGF, IGFBP3, ICAM-2 and ICAM-3 genes differentially expressed by DECs and ADMECs. The data represent the mean ± SD of triplicate samples from five separate experiments, * *p* < 0.05, ** *p* < 0.01, *** *p* < 0.005. (**C**) Evaluation of the production of VEGF-A, HGF and IGFBP3 proteins in the supernatants of a confluent monolayer of DECs and ADMECs after 4 h of culture using a commercial ELISA kit. The data represent the mean ± SD of triplicate samples from five separate experiments. * *p* < 0.05, ** *p* < 0.01, *** *p* < 0.005. (**D**) Cytofluorimetric analysis for the expression of ICAM-2 and ICAM-3 in basal condition of freshly isolated DECs and ADMECs. The ECs were incubated with PE-conjugated mouse anti-human ICAM-2 and ICAM-3 mAb. PE-conjugated isotype-matched IgG2a or IgG1 were used as negative control, respectively. Data are represented as mean ± SD of the Mean Fluorescence Intensity (MFI) of five separate experiments.

**Figure 2 ijms-20-01604-f002:**
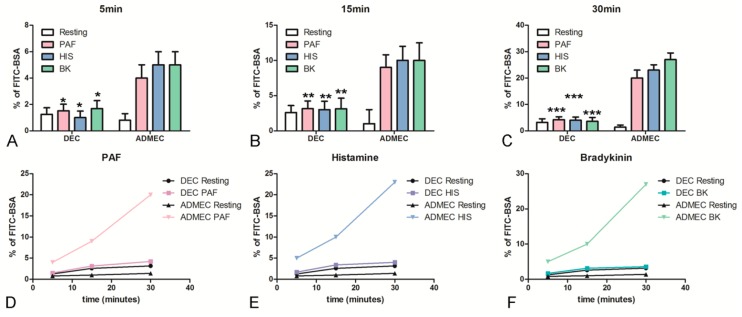
Permeabilizing activity of endothelial cells to classical vasoactive stimuli. The permeabilizing activity was evaluated kinetically, after 5 (**A**), 15 (**B**) and 30 min (**C**) adding PAF (**D**), HIS (**E**) or BK (**F**), to the upper chamber of the TW, measuring the amount of FITC-labeled BSA that leaked through a monolayer of endothelial cells into the lower chamber. The data represent the mean ± SD of duplicate samples from four separate experiments * *p* < 0.05; ** *p*<0.01; *** *p* < 0.005.

**Figure 3 ijms-20-01604-f003:**
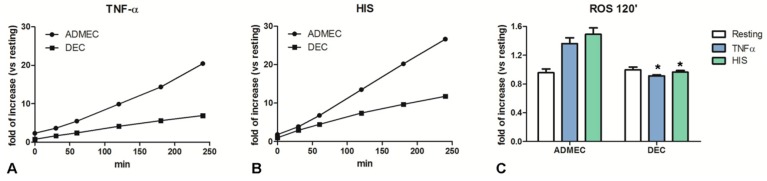
Production of intracellular ROS by endothelial cells. ROS production by endothelial cells exposed to TNF-α (**A**) or HIS (**B**). Following pre-treatment with TNF-α (100 ng/mL) or histamine (HIS; 0.1 µM), ECs were stained with Ampliflu Red for the evaluation of H_2_O_2_ production after 30, 60 120, 180 and 240 min. (**C**) Histograms represent the production of intracellular ROS by endothelial cells stimulated with TNF-α or HIS after 120 min. The data represent the mean ± SD of triplicate samples from five separate experiments; * *p* < 0.05.

**Figure 4 ijms-20-01604-f004:**
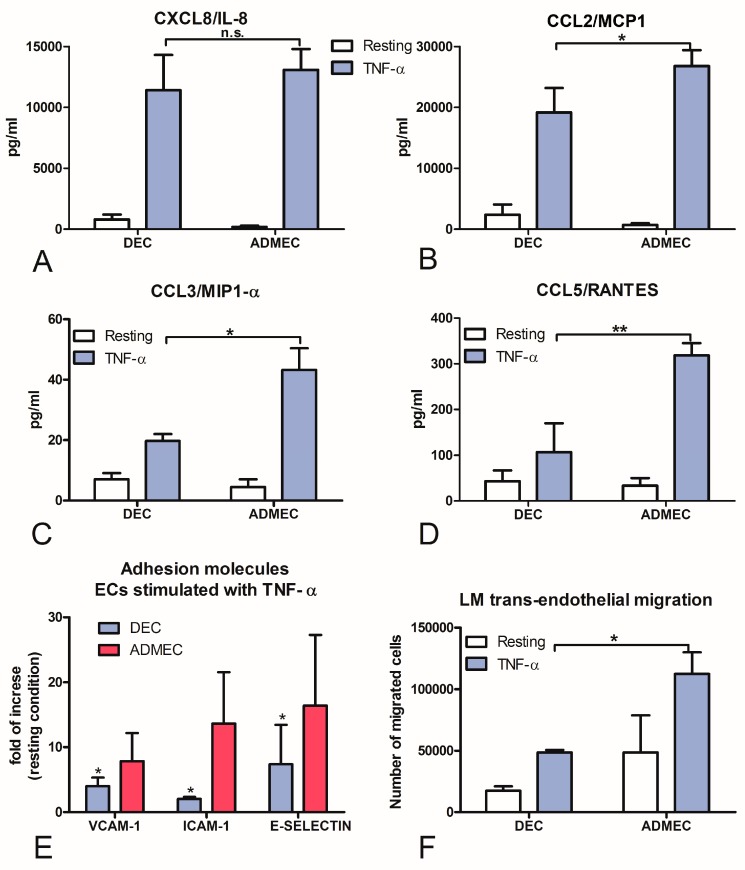
Secretion of chemokines and surface-expression of adhesion molecules by ECs stimulated with TNF-α. CXCL8/IL-8 (**A**), CCL2/MCP-1 (**B**), CCL3/MIP-1α (**C**) and CCL5/RANTES (**D**) production in DEC and ADMEC supernatants after 4 h incubation with TNF-α was measured using a beads-based multiplex immunoassay (Luminex^®^). (**E**) ELISA on the whole cells for the expression of ICAM-1, VCAM-1 or E-Selectin on DEC and ADMEC plasma membrane after 4 h or 18 h incubation with TNF-α. The data represent the mean ± SD of triplicate samples from five separate experiments * *p* < 0.01; ** *p* < 0.005. (**F**) Trans-endothelial migration of Lympho-Monocytes (LM) across untreated and TNF-α-treated DEC and ADMEC: migration is shown as the number of migrated cells. The data represent the mean ± SD of triplicate samples from three separate experiments * *p* < 0.05; n.s. not significant.

**Figure 5 ijms-20-01604-f005:**
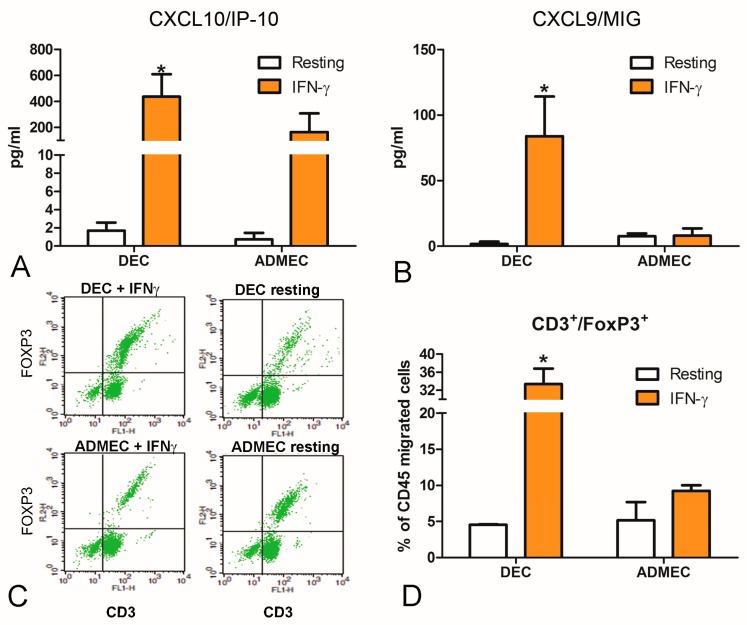
Secretion of chemokines and trans-endothelial migration of LM through EC stimulated with IFN-γ. The production of CXCL10/IP-10 (**A**) and CXCL9/MIG (**B**) in DEC and ADMEC supernatants after 4 h incubation with IFN-γ was measured using a beads-based multiplex immunoassay (Luminex^®^). The data represent the mean ± SD of triplicate samples from five separate experiments * *p* < 0.01. (**C**,**D**) Trans-endothelial migration of LM across untreated and IFN-γ-treated DEC and ADMEC. (**C**) Representative dot plots for flow cytometry of trans-endothelial migrated LM, stained for CD3 and FoxP3. (**D**) Quantitation of the percent of total migrated LM cells, positive for both CD3 and FoxP3 (CD3^+^FoxP3^+^) by flow cytometry. Migration was presented as the percentage of CD45 migrated cells. The data represent the mean ± SD of duplicate samples from three separate experiments * *p* < 0.05.

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
