# Peer review of "Transcriptomics and Immunological Analyses Reveal a Pro-Angiogenic and Anti-Inflammatory Phenotype for Decidual Endothelial Cells"

_ijms, 2019, doi:10.3390/ijms20071604_

Round 1

Reviewer 1 Report

Agostinis and colleagues have obtained endothelial cells from decidual and skin biopsies of patients. The endothelial cultures were compared by cellular and molecular tests, and showed differences with respect to response to cytokine (TNFa, IFNg) stimulation, permeability to inflammatory cells, and H2O2 production.

The manuscript is well structured, and it has clearly described methods and a logical build-up of results. It does contain some language and writing errors. I have a few comments:

The second part of the Title is inappropriate and too speculative, since EC1/EC2 is not mentioned at all until the last line of the Conclusions and no results are referring to this distinction.

The manuscript is unclear about the number of women included for EC isolation and culturing. These numbers should be included. In some cases, such as Fig 1A and Fig 4, the number of separate experiments is mentioned, but does this mean the number of patients or different cell passages from the same culture/patient?

Results for the cytofluorimetric analyses are not convincing. Fig. 1D only shows an n=1 experiment, where the isotype controls for DECs and ADMECs are not at the same level.

In the Results section it is mentioned that microarray experiments were not shown, but actually there is a Suppl Fig 1, which is not referred to in the text.

Fig. 5D: what were the total number of cells that migrated? For TNFa (Fig. 4F), the DECs were about 2.5 fold lower than ADMECs, which could “dilute” the effect seen for increased %Foxp3 within CD45 total.

The statement in the Discussion section that DECs failed to upregulate several chemokines upon TNFa is questionable, since Fig. 4 shows considerable fold differences between resting ansd TNFa. I suggest it to be removed.

Author Response

We are grateful to the Editor and the reviewers for their comments.

We are submitting the revised version of the manuscript regarding the reviewer’s suggestions and language editing.

POINT BY POINT REPLY

REV 1

The second part of the Title is inappropriate and too speculative, since EC1/EC2 is not mentioned at all until the last line of the Conclusions and no results are referring to this distinction.

We do not completely agree with the reviewer’s comment. Anyway, the title has been modified and revised as requested.

The manuscript is unclear about the number of women included for EC isolation and culturing. These numbers should be included. In some cases, such as Fig 1A and Fig 4, the number of separate experiments is mentioned, but does this mean the number of patients or different cell passages from the same culture/patient?

Each population refers to a single patient. This information is now indicated in the M&M section.

Results for the cytofluorimetric analyses are not convincing. Fig. 1D only shows an n=1 experiment, where the isotype controls for DECs and ADMECs are not at the same level.

We agree with the reviewer and thank him for his right observation. Now, in figure 1 we converted the cytofluorimetric data in histrograms showing the mean (+/- standard error) of the MFI deriving from different experiments.

In the Results section it is mentioned that microarray experiments were not shown, but actually there is a Suppl Fig 1, which is not referred to in the text.

We apologise to the referee for this mistake, now we added the indication of ‘Suppl Fig 1’ as required.

Fig. 5D: what were the total number of cells that migrated? For TNFa (Fig. 4F), the DECs were about 2.5 fold lower than ADMECs, which could “dilute” the effect seen for increased %Foxp3 within CD45 total.

We thank the reviewer for this proper observation. The total number of cells migrated with IFN-γ is not significantly different between the two populations. For this reason we are confident that the effect seen for increased % of Foxp3 within CD45 total is not affected by “dilution”.

The statement in the Discussion section that DECs failed to upregulate several chemokines upon TNFa is questionable, since Fig. 4 shows considerable fold differences between resting ansd TNFa. I suggest it to be removed.

We agree with this comment and, as suggested by the reviewer, we removed the statement from the Discussion section.

Reviewer 2 Report

This study is designed to investigate the phenotype of decidual endothelial cells (DECs) by comparison with adult dermal microvascular endothelial cells (ADMECs).  The authors have demonstrated that DECs enhance angiogenesis-related genes expression.  On the other, DECs have reduced permeabilizing activity, ROS production, and chemokines expression compared to ADMECs.  Moreover, the authors have shown that DECs might predominantly induce NK cells and Treg recruitment.  The findings of this study are very interesting.  However, this reviewer believes that the manuscript requires some changes in order to improve its quality.

Title should be clear and reflecting the aim, approach, and results of the study.

2. The paper uses many ambiguous or unnecessary abbreviations.

3. It is important to provide a more detailed description of the specific properties of DECs and ADMECs, such as molecule makers and difference to other ECs, in the introduction.

4. It is not clear the rationale for comparison with ADMECs. What is the reason of comparison to ADMECs.

5. Fluorescence images of VWF and vimentin is not clear, because there are no controls.  Western blot might be better to indicate the expression of them.

6. What is the mean of ‘five different populations of DECs/ADMECs (L111)’.  Are there any different phenotypes in each cell?

7. Figure 1D, MFI might be better than histogram.  Statistical difference should be shown.

8. Figure 2 and 3, two type of figures based on same data is unnecessary.  In addition, the authors need to clarify the comparison targets.

9. Figure 4 A and B are similar to reported data in previous authors paper in Scientific reports. 

10. Generally, it is difficult to compare different cells.  It is not clear the rationale for the comparison; for example, normalization between cells and difference of internal control expression in cells.

11. In Figure 4E, ELISA can detect soluble form, membrane bind, intracellular form of molecules.  In order to discuss about cell surface form, flow cytometric analysis is better.

12. Figure5, Data of NK cells are needed.  A more detailed description of the result and experimental condition by showing appropriate control of selective transmigration should be provided.

13. Discussion section only is rehash of results.  The authors should describe summary of results, the importance of this study, and strong points of this study, in the context of EC characterization, angiogenesis, trans-endothelial migration, and pro-/anti-inflammation.

14. The concept of EC1/EC2 phenotype might be interesting, but it is too speculative.  In this study, the authors have not demonstrated the definition, evidence, and supportive data, the authors should remove this concept.

Author Response

We are grateful to the Editor and the reviewers for their comments.

We are submitting the revised version of the manuscript regarding the reviewer’s suggestions and language editing.

POINT BY POINT REPLY

REV2

This study is designed to investigate the phenotype of decidual endothelial cells (DECs) by comparison with adult dermal microvascular endothelial cells (ADMECs). The authors have demonstrated that DECs enhance angiogenesis-related genes expression. On the other, DECs have reduced permeabilizing activity, ROS production, and chemokines expression compared to ADMECs. Moreover, the authors have shown that DECs might predominantly induce NK cells and Treg recruitment. The findings of this study are very interesting. However, this reviewer believes that the manuscript requires some changes in order to improve its quality.

Title should be clear and reflecting the aim, approach, and results of the study.

2. The paper uses many ambiguous or unnecessary abbreviations.

We removed and modified some ambiguous abbreviations.

3. It is important to provide a more detailed description of the specific properties of DECs and ADMECs, such as molecule makers and difference to other ECs, in the introduction.

Now it has been included in Introduction section.

4. It is not clear the rationale for comparison with ADMECs. What is the reason of comparison to ADMECs.

Skin is the primary interface between the body and the external environment and skin injury or infection promptly results in the activation of an efficient inflammatory response. Dermal microvascular endothelial cells are active participants and regulators of the inflammatory process at a site of inflammation [5]. They contribute to the process secreting inflammatory mediators, bactericidal molecules, increasing leakiness and pro-coagulant activity, modulating adhesion and transmigration of leukocytes through the expression of adhesion molecules and chemokines and increasing pro-inflammatory cytokines expression [6]. On the contrary, the decidua is a very particular tissue from the immunological point of view. It is the place where maternal-fetal tolerance is established.

5. Fluorescence images of VWF and vimentin is not clear, because there are no controls. Western blot might be better to indicate the expression of them.

The characterization of endothelial cells is based not only on the quantity of protein expression but also on the molecular pattern of the antigens. For istance, vWF in endothelial cells has a typical dotted pattern while vimentin is widely diffused as filaments along all the cytoplasm and not in the nucleus. For this reason the control of this kind of analysis is the specificity of the pattern.

6. What is the mean of ‘five different populations of DECs/ADMECs (L111)’. Are there any different phenotypes in each cell?

Each population refers to a single patient. The variability of primary cells is higher than that observed in cell lines and better represents the variability among patients. This information is now indicated in the M&M section.

7. Figure 1D, MFI might be better than histogram. Statistical difference should be shown.

Now, in figure 1 we converted the cytofluorimetric data in histrograms showing the mean (+/- standard error) of the MFI.

8. Figure 2 and 3, two type of figures based on same data is unnecessary. In addition, the authors need to clarify the comparison targets.

We agree with the reviewer that the two figures are based on the same set of data. However, we prefer to maintain both figures because, as presented, they improve the clarity of our findings.

9. Figure 4 A and B are similar to reported data in previous authors paper in Scientific reports.

In Masat et al. ‘Scientific Reports’ we evaluated the response of DECs and ADMECs to LPS, an exogenous stimulus that plays a major role only during bacterial infection; in the present study, we evaluated and compared the response of these endothelial cells to pro-inflammatory stimuli (TNF-α and IFN-γ).

10. Generally, it is difficult to compare different cells. It is not clear the rationale for the comparison; for example, normalization between cells and difference of internal control expression in cells.

It is not the case of endothelial cells because they show the same morphology and physiologically growth to confluence covering the bottom of the plate wells. In any case, for each quantification experiment, we checked the homogeneity of the wells evaluating the total protein concentration by Bradford assay; whereas for RT-qPCR the normalization was performed analyzing 18S expression. Having observated that replication rate of the population was exactly the same, we are confident that, seeding the same number of cells on to the TW, we are working in identical condition also for migration and permeability assays.

11. In Figure 4E, ELISA can detect soluble form, membrane bind, intracellular form of molecules. In order to discuss about cell surface form, flow cytometric analysis is better.

We don’t agree with the reviewer’s observation. As he surely knows, endothelial cells grow in adhesion and, therefore, they have to be detached from the wells by means of a Trypsin/EDTA treatment and resuspended to perform the cytofluorimetric analysis. During such enzymatic treatment, digestion, some antigens expressed on the cell surface may be truncated or even removed, thereby preventing their recognition by the specific antibody. Consequently, the obtained specimen could be recorded as a false negative. Furthermore, endothelial cells behave differently from other cell types, such as leukocytes. They physiologically grow and live attached to matrixes and to other cells forming a complete monolayer along the inner face of vessels differenciating an apical and a luminal side. Therefore, is our opinion that ELISA performed on whole cells still attached to the well surface represents the best technique to analyze antigen expressed onto the cell side that is physiologically exposed to the blood stream.

12. Figure5, Data of NK cells are needed. A more detailed description of the result and experimental condition by showing appropriate control of selective transmigration should be provided.

According to the reviewer’s suggestion, these data are shown in Supplemental Figure 3 of the revised version of the paper.

13. Discussion section only is rehash of results. The authors should describe summary of results, the importance of this study, and strong points of this study, in the context of EC characterization, angiogenesis, trans-endothelial migration, and pro-/anti-inflammation.

To meet the reviewer’s request, the Discussion section has been modified and revised.

14. The concept of EC1/EC2 phenotype might be interesting, but it is too speculative. In this study, the authors have not demonstrated the definition, evidence, and supportive data, the authors should remove this concept.

To meet the reviewer’s request, the title has been modified and revised.

Reviewer 3 Report

This comparison undertaken by Agostinis et al. of selected features of DECs and ADMECs found that DECs produce higher levels of HGF, VEGF-A, IGFBP3, ICAM2 and ICAM3 expression, and are capable of modulating leukocyte recruitment. Overall, the data presented are quite clearly set out. The conclusions are well supported by the results. However, the present form of this work contains a few drawbacks that need be addressed, as specified below:

1. The results of the microarray (GEP) data are important and should be provided in MS or supplement (lines 89, 106).

2. The results of Figure 1D would be better quantified in a bar graph, so that readers can better understand the results.

3. In Figure 4E, the variability of data (error bar) is too large, so the authors need to undertake a series of separate experiments to prevent statistically inconclusive results.

4. Commonly used markers for T-reg identification include CD25, FoxP3, and CD127, but these authors only used FoxP3. This experiment therefore needs to be expanded upon with data from CD25 and CD127 findings.

5. In the Materials and Methods section, the isolation of lymphocytes from monocytes is missing and should be added. The Results section (line 174) mentions the isolation of lymphocytes from monocytes, without giving the findings.

6. Extensive English editing must be performed by native English speaker or professional company.

Author Response

We are grateful to the Editor and the reviewers for their comments.

We are submitting the revised version of the manuscript regarding the reviewer’s suggestions and language editing.

POINT BY POINT REPLY

REV3

1. The results of the microarray (GEP) data are important and should be provided in MS or supplement (lines 89, 106).

The data are now shown in the Supplemental Figure 1.

2. The results of Figure 1D would be better quantified in a bar graph, so that readers can better understand the results.

We agree with the reviewer’s suggestion. Now, in Figure 1 we converted the cytofluorimetric data in histrograms showing the mean (+/- standard error) of the MFI.

3. In Figure 4E, the variability of data (error bar) is too large, so the authors need to undertake a series of separate experiments to prevent statistically inconclusive results.

We agree with the observation of the referee but the variability of the response of primary cells is in general very high. but in each experiment performed, the expression of the adhesion molecules significantly differed between DEC and ADMEC.

4. Commonly used markers for T-reg identification include CD25, FoxP3, and CD127, but these authors only used FoxP3. This experiment therefore needs to be expanded upon with data from CD25 and CD127 findings.

The referee’s observation is undoubtely appropriate. In this respect, additional experiments will be run as to further characterize, at molecular level, Treg cells and their interaction with DEC during the recruitment in human decidua. The results of such investigations will be hopefully included in a forthcoming paper. A sentence recalling the usefulness of a further characterization of Treg markers is included in the discussion section of the revised version of the paper.

5. In the Materials and Methods section, the isolation of lymphocytes from monocytes is missing and should be added. The Results section (line 174) mentions the isolation of lymphocytes from monocytes, without giving the findings.

In our experiments we used the lympho-monocyte population (LM). We do not separated the two population before transendothelial migration

6. Extensive English editing must be performed by native English speaker or professional company. As requested, the paper has been carefully revised.

Round 2

Reviewer 1 Report

I suggest to add information and reference(s) on EC2 in the Discussion section, since it is now still only mentioned in the last sentence of the conclusion.

Apart from that, the comments have been adequately addressed by the authors.

Author Response

We are grateful to the Editor and the reviewers for their comments.

We are submitting the revised version of the manuscript regarding the reviewer’s suggestions and language editing. We changed again the title trying to improve the clearness and the sound. Hopefully we have found favor with referees.

POINT BY POINT REPLY

Rev1

I suggest to add information and reference(s) on EC2 in the Discussion section, since it is now still only mentioned in the last sentence of the conclusion.

Following the request of the reviewer 1, the description of the hypotetical difference among EC1 and EC2 has been removed.

Apart from that, the comments have been adequately addressed by the authors.

Reviewer 2 Report

The revised manuscript has been improved. However, this reviewer believes that the manuscript requires some changes.

1. Still needs to check abbreviations such as TW, ICAM2 or ICAM-2.  In abbreviations section, the authors have used some different style and should check abbreviations of ADMECs and others.

2. In figure1A, change the order, DEC is better to show in the upper.

3. Figure 5, this transmigration assay is very valuable. The authors should describe more detail about this assay including actual example in previous literatures or control experiments.

4. The authors should remove EC1 or EC2 in abstract (L36), introduction (L84, L85), conclusion (L419), and title of supplemental material.

Author Response

We are grateful to the Editor and the reviewers for their comments.

We are submitting the revised version of the manuscript regarding the reviewer’s suggestions and language editing. We changed again the title trying to improve the clearness and the sound. Hopefully we have found favor with referees.

POINT BY POINT REPLY

Rev2

The revised manuscript has been improved. However, this reviewer believes that the manuscript requires some changes.

1. Still needs to check abbreviations such as TW, ICAM2 or ICAM-2.  In abbreviations section, the authors have used some different style and should check abbreviations of ADMECs and others.

The abbreviations have been checked.

2. In figure1A, change the order, DEC is better to show in the upper.

Following your the suggestion the Figure has been changed.

3. Figure 5, this transmigration assay is very valuable. The authors should describe more detail about this assay including actual example in previous literatures or control experiments.

The previous paper concerning the migration assay has now been included in the material end method section.

4. The authors should remove EC1 or EC2 in abstract (L36), introduction (L84, L85), conclusion (L419), and title of supplemental material.

Done.